# New suspension-feeding radiodont suggests evolution of microplanktivory in Cambrian macronekton

Rudy Lerosey-Aubril [1] & Stephen Pates [2,3]

The rapid diversification of metazoans and their organisation in modern-style marine ecosystems during the Cambrian profoundly transformed the biosphere. What initially sparked this Cambrian explosion remains passionately debated, but the establishment of a coupling between pelagic and benthic realms, a key characteristic of modern-day oceans, might represent a primary ecological cause. By allowing the transfer of biomass and energy from the euphotic zone—the locus of primary production—to the sea floor, this biological pump would have boosted diversification within the emerging metazoan-dominated benthic communities. However, little is known about Cambrian pelagic organisms and their trophic interactions. Here we describe a filter-feeding Cambrian radiodont exhibiting morphological characters that likely enabled the capture of microplankton-sized particles, including large phytoplankton. This description of a large free-swimming suspension-feeder potentially engaged in primary consumption suggests a more direct involvement of nekton in the establishment of an oceanic pelagic-benthic coupling in the Cambrian.

[1] Palaeoscience Research Centre, School of Environmental and Rural Science, University of New England, Armidale, NSW 2351, Australia. [2] Department of Zoology, University of Oxford, South Parks Road, Oxford OX1 3PS, UK. [3] Institute of Earth Sciences, University of Lausanne, Lausanne CH-1015, Switzerland. Correspondence and requests for materials should be addressed to R.L-A. (email: leroseyaubril@gmail.com)

The evolution of phytoplanktonic life dates back to at least 1.8 Ga, but it remained essentially represented by simple forms of acritarchs (leiosphaerids) until the end of the Proterozoic Eon[1]. During this time interval, eukaryotic life evolved significantly, as evidenced by the emergence of large unicellular forms[2] (also see ref. [3]) or the first macroscopic multicellular organisms[4], including early animals[5,6]. However, these important steps of eukaryotic evolution were apparently confined to the benthic realm, which remained essentially decoupled from primary production in the euphotic zone of the oceans[1]. The early Cambrian Period is associated with an even greater modification of marine life—the advent of metazoans and their organisation in complex ecosystems. In less than 30 Myr, most major bilaterian phyla appeared and greatly diversified[7,8], provoking a dramatic change in both the composition and the functioning of the biosphere, which in turn profoundly and irreversibly impacted the Earth system[9,10]. For instance, likely triggered by a new ecological force—predation —some of these early Cambrian animals (e.g. scalidophorans) rapidly evolved aptitudes for life within soft substrates[11], thus increasing the breadth and depth of eukaryotic colonization of the sea floor, and concomitantly the habitability of the latter[12]. Interestingly, phytoplankton also radically changed at that time with the radiation of small, rapidly evolving ornamented acritarchs (acanthomorphs). This has been interpreted as indirect evidence of the introduction of a key component of modern pelagic ecosystems— herbivorous zooplankton[13]. These primary consumers play a critical role in the transfer of energy and biomass from the well-lit surface layer down to the sea floor, repackaging organic matter produced by photosynthesis into larger, more rapidly sinking elements (e.g. carcasses, faecal pellets). The presence of early Cambrian herbivorous zooplankton was demonstrated by the finding of microscopic carbonaceous remains of crustacean filter-feeding apparatuses[14,15]. Similar, but larger fossils were also recovered from middle Cambrian strata of the Deadwood Formation in western Canada, which also yielded middle–upper Cambrian remains of omnivorous zooplanktonic crustaceans[16]. Although rare, these fossils attest to a complexification of pelagic ecosystems and the onset of a coupling between pelagic and benthic realms in the Cambrian.

Until recently, evidence for a significant contribution of macroscopic animals to the biological pump at that time was meagre. The few nektonic taxa (e.g. chaetognaths, a few arthropods) found in Burgess Shale-type deposits were thought to predominantly live close to the sea floor (demersal zone[17]). The description of a lower Cambrian, filter-feeding representative of Radiodonta—distant relatives of spiders, insects, myriapods and crustaceans previously regarded as fearsome apex-predators—radically changed this picture[18]. Comparing radiodonts to pachycormid fish, sharks and whales, Vinther et al.[18] concluded that the evolution of suspension-feeding was somewhat predictable in these groups, the ancestors combining predatory habits and large body sizes. This assumption found tremendous support in the discovery of a two metre-long, suspension-feeding radiodont from the Lower Ordovician Fezouata Shale of Morocco[19].

In this contribution, we show that *Pahvantia hastata*, an as-yet enigmatic arthropod from the middle Cambrian of Utah, was also a suspension-feeding radiodont, although a relatively small one. This taxon illustrates that shortly after the Cambrian explosion, some nektonic animals were likely capable of feeding on microplankton, including large phytoplankton, and therefore of contributing to primary consumption and the establishment of the biological pump in the oceans.

## Results

**Morphological characteristics**. *Pahvantia hastata* is a relatively small radiodont (estimated total body length <25 cm), occurring exclusively in the fine-grained clastic outer-shelf deposits of the Drumian Wheeler Formation, in both the House Range and the Drum Mountains of western Utah (Supplementary Fig. 1 and ref. [20]). This taxon was previously known from four specimens assigned to two different arthropods[20,21]. Three were initially interpreted as univalve carapaces of *P. hastata*, at that time an arthropod of uncertain affinities. The fourth was originally regarded as the isolated valve of a bivalved carapace belonging to *Proboscicaris agnosta*; it was recently identified as a lateral element of a radiodont cephalic carapace and re-assigned to the genus *Hurdia*[22,23]. A new fossil, composed of a dislocated cephalic carapace associated with a frontal appendage (Fig. 1), demonstrates that the aforementioned specimens belong to a single species of hurdiid radiodonts.

Like other members of this family (*Aegirocassis*[19] and *Hurdia*[22,24]), *Pahvantia* possesses a well-developed tripartite cephalic carapace (Fig. 1, Supplementary Fig. 2). Its central element exhibits a particularly distinctive morphology, characterised by a lanceolate outline, a tiny antero-sagittal spine, well-developed lateral lips, a strong postero-lateral constriction, and an indented posterior margin. It is associated with a pair of tall lateral elements, with slightly concave posterior margins and possibly hook-shaped, anterior processes. This association is illustrated by specimen KUMIP314089, in which the three carapace plates are of comparable sizes and still in contact anteriorly, as if the dorsal element had tilted forwards relative to the lateral ones, but had remained attached to them at its anterior tip (Fig. 1a, b).

The same specimen provides unequivocal evidence for the radiodontan affinities of *Pahvantia* in the form of a frontal appendage projecting from under the left side of the dorsal element (Fig. 1a). The organisation of this appendage is imperfectly understood due to preservation, but also the superimposition of many elements, especially a multitude of long hair-like structures tentatively interpreted as setae hereafter. The peduncle (proximal part) is particularly narrow proximally and widens distally (Fig. 1c, e), possibly indicating an oblique orientation relative to bedding and/or incomplete preservation. Lateral constrictions associated with lighter-coloured transverse lines suggest that it may comprise up to five podomeres, but this aspect remains unclear. More distally, the appendage is essentially represented by endites of two, radically different types. The two proximal ones are short and robust plate-like structures, each bearing seven anterior auxiliary spines, apparently organised in two rows (Fig. 1c, e, f). The more distal of these robust endites is twice the width of the other and inserts more distally on the appendage, showing that the two endites are not forming a pair. They are followed distally by five, apparently unpaired endites that are two to three times wider and about three times longer, and exhibit anterior margins fringed by numerous setae (Fig. 1c–e). Each long endite appears to branch into two parts towards the tip, located on slightly different topographic planes. We believe that this apparent split actually results from the offsetting distally of two rows of setae born by each endite (Supplementary Fig. 3, Supplementary Note 1). Each row is composed of 50–60 setae 100–150 μm wide at base, which are evenly spaced 60–70 μm apart and arranged subparallel to one another and to the setae of other rows. The proximal setae of each row are long (up to 3.7 mm), typically overlapping the endite, if not the two endites distal to the one bearing them (Fig. 1d). Setae towards the tip of the endite are represented by their basal parts only, which may indicate that they were more delicate (Fig. 1c, e). On the other hand, this part of the fossil is associated with a coarser-grained and harder matrix, which may explain this incomplete preservation of such fragile structures.

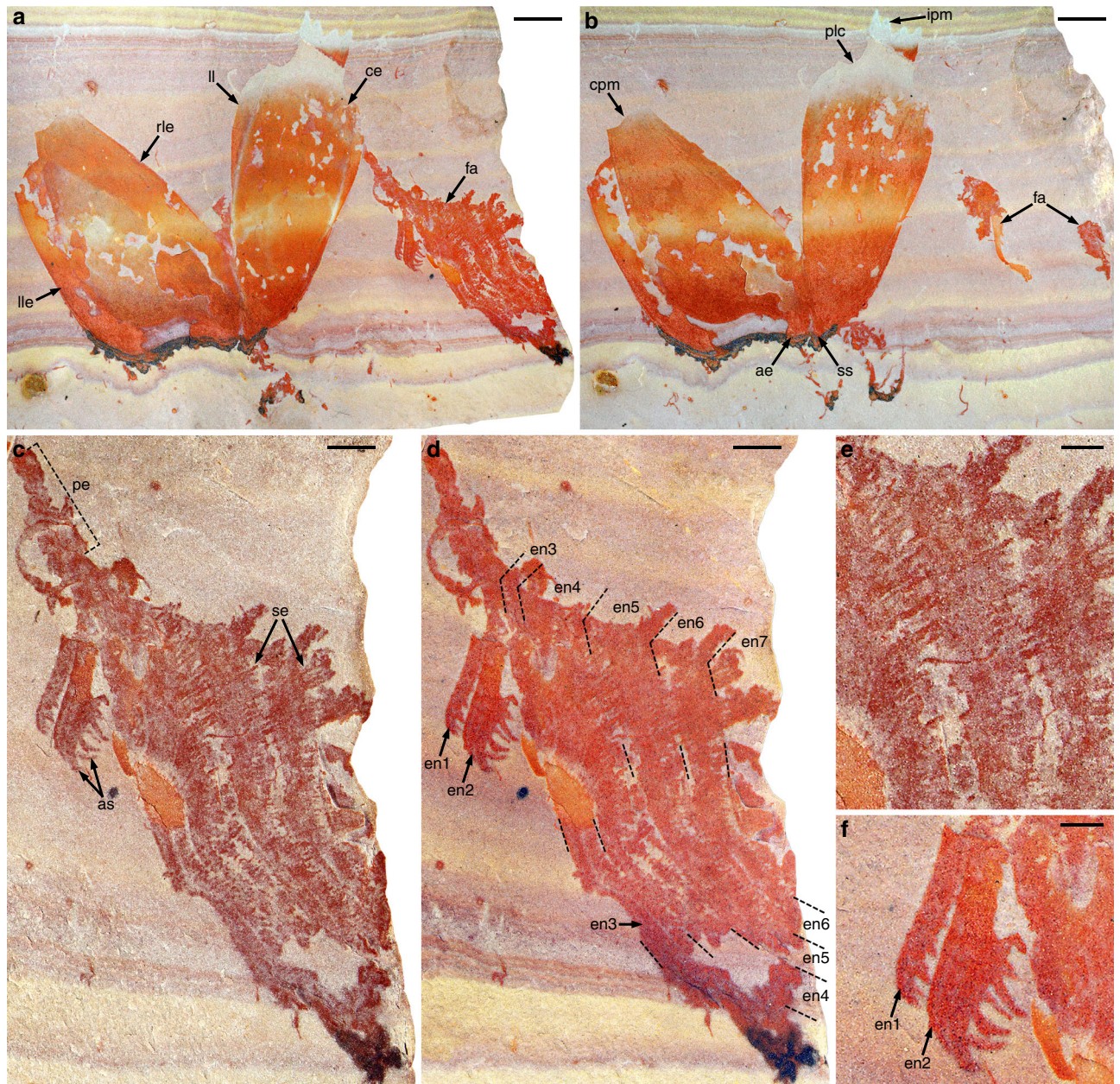

**Fig. 1** Assemblage of cephalic elements of *Pahvantia hastata*. Photographs of part and counterpart of specimen KUMIP314089 immersed in dilute ethanol (**a**, **b**, **d**, **f**) or dry (**c**, **e**). **a**, **b** General views of part (**a**) and counterpart (**b**). **c–f** Details of the frontal appendage (**c**, **d**), its setae (**e**), and its proximalmost two endites (**f**). Scale bars represent 5 mm (**a**, **b**), 2 mm (**c**, **d**), 1 mm (**e**, **f**). ae anterior extension, as auxiliary spines, ce central element, cpm concave posterior margin, en1–7 endites 1–7, fa frontal appendage, ipm indented posterior margin, ll lateral lip, lle left lateral element, pe peduncle, plc postero-lateral constriction, rle right lateral element, se setae, ss anterior sagittal spine

The new material also includes two isolated dorsal elements (Supplementary Fig. 2e, f), which further illustrate the marked distinctiveness of this skeletal element in *Pahvantia hastata*. Interestingly, one of these specimens has not been found in the House Range, but in the Drum Mountains (Supplementary Fig. 2e). A more detailed description of *Pahvantia hastata* and a revision of its systematics are provided in the Supplementary Note 1, along with a formal diagnosis of the family Hurdiidae.

**Phylogenetic relationships**. To explore the phylogenetic position of *Pahvantia hastata* and the impact of its characters on the phylogeny of radiodonts, a parsimony analysis of 61 morphological characters scored for 33 taxa was performed

(Supplementary Note 2 and ref.[20]). Under equal weighting, this analysis yielded 144 most parsimonious trees (MPTs) of 104 steps (CI = 0.621, RI = 0.804; Fig. 2). Except for *Caryosyntrips*, radiodonts constitute a monophyletic clade supported by the presence of an oral cone with differently-sized plates, auxiliary spines on the protocerebral appendages, and trunk setal blades/exites extending to the dorsal region. This clade is sister to deuteropods and consists of two subgroups—anomalocaridids and amplectobeluids on the one hand, and tamisiocaridids[25] (formerly Cetiocaridae[18]) and hurdiids on the other. Both are defined by characteristics of the endites, their alternation in size for the former and their elongate plate-like shape for the latter. This topology is largely consistent with

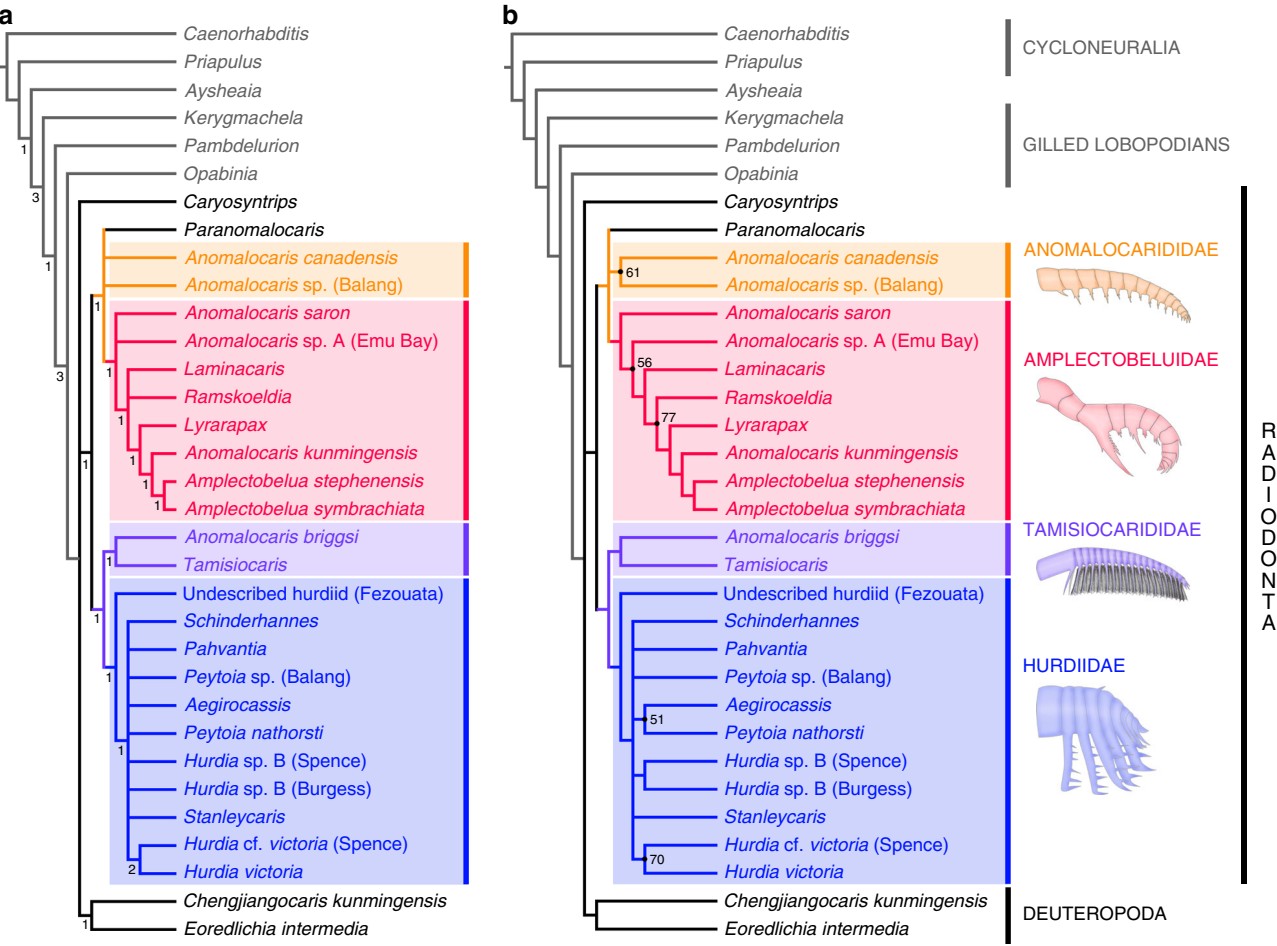

**Fig. 2** Phylogenetic position of *Pahvantia hastata*. **a** Strict consensus of 144 most parsimonious trees (MPTs) at 104 steps under equal weighting (CI = 0.621, RI = 0.804), which shows that *P. hastata* belongs to the family Hurdiidae. Numbers at the nodes indicate Bremer support values. **b** The 50% majority-rule consensus of 144 MPTs at 104 steps under equal weighting (CI = 0.661; RI = 0.836). Numbers at the nodes indicate percentage of trees containing that node, and nodes without numbers were recovered in all trees. The same topology is recovered under implied weighting and different values (3, 4, 9, 10) of concavity constant *k*

those of previously published trees[18,19,26,27], despite major changes to the original matrix of Vinther et al.[18] (Supplementary Note 2).

Whether the analysis is performed under equal or implied (*k* = 1–10) weighting, *Pahvantia* is constantly retrieved within the Hurdiidae in a polytomy with most members of this family (Fig. 2), including the giant suspension-feeder *Aegirocassis*. However, the latter taxon forms a clade with the sediment-sifter *Peytoia nathorsti* in 51 (*k* = 3) to 56 (*k* = 2, 10) percent of the trees (Fig. 2b), which might indicate that its specialized feeding mode might have evolved independently from the *Pahvantia* lineage. The other two putative filter-feeding radiodonts, *Anomalocaris briggsi* and *Tamisiocaris borealis*, are usually recovered as a monophyletic family Tamisiocarididae, which is sister to hurdiids and supported by a slight retrocurvature of all endites.

Running the analysis under implied weighting (*k* = 1–10) yields tree topologies virtually identical to that obtained under equal weighting (Supplementary Fig. 4). One exception concerns *A. briggsi*, which is retrieved as the sister-taxon of all anomalocaridids and amplectobeluids when using low concavity constants (*k* = 1, 2). The second is more common (*k* = 2, 5–8) and corresponds to the grouping of *Stanleycaris hirpex* and *Hurdia* sp. B from the Burgess Shale and Spence Shale Konservat-Lagerstätten. Although never retrieved in more than 54 percent of the trees (*k* = 8) in our analysis, this

association has been repeatedly observed in the past[19,26,27] and may therefore truly indicate close affinities between these taxa. This great coherence between the results obtained using equal and implied weighting gives credence to the phylogenetic models presented in Fig. 2.

The same dataset was also subjected to a Bayesian inference analysis (Supplementary Fig. 5), which recovers *Pahvantia* as part of a monophyletic Hurdiidae. Tamisiocarididae and the core group Amplectobeluidae (*Amplectobelua*, *Lyrarapax* and *Anomalocaris kunmingensis*) also form monophyletic groups. All the other radiodonts are retrieved in polytomy with these three clades and a monophyletic Deuteropoda, except for *Caryosyntrips* that occupies a more basal position. Despite a general loss of resolution inherent to Bayesian inference analyses[28], the topology of this tree is fundamentally congruent to those obtained using parsimony analyses (Supplementary Figure 4).

Whatever the methods used, the resulting trees clearly illustrate two important points: (1) *Pahvantia* belongs to the Hurdiidae and (2) suspension-feeding evolved in at least two independent lineages of radiodonts (tamisiocaridids and hurdiids).

**Discussion**

Although imperfectly understood, the morphology of *Pahvantia* frontal appendages strongly suggests that they were used for the

capture of microscopic food particles suspended in the water column. The number (>50 per row), morphology (thin and long), and distribution (even and dense) of the putative setae indicate that they formed a filtering apparatus (Fig. 1c–e), rather than manipulated moving macroscopic preys. Except for the proximal endites (Fig. 1f), likely used to comb the filtering setae and transfer food particles to the mouth, the frontal appendage of *Pahvantia* is devoid of the robust spines (endites and auxiliary spines) characterizing radiodont appendages with inferred grasping (e.g. *Anomalocaris*, *Amplectobelua*[29,30]), grasping-crushing (e.g. *Amplectobelua*[31]), grasping-slicing (e.g. *Caryosyntrips*, *Lyrarapax*[27,32]), or sediment sifting (e.g. *Hurdia*, *Peytoia*[23]) functions.

Suspension-feeding habits have been inferred for two, possibly three radiodonts exhibiting fundamentally different organisation of the frontal appendages. In the early Cambrian *Tamisiocaris*, and possibly *Anomalocaris briggsi*, suspended particles (including mesoplankton) were trapped by a net formed by the numerous, needle-like auxiliary spines fringing the anterior and posterior margins of long and delicate endites[18] (Fig. 3a, b). In contrast, the Early Ordovician *Aegirocassis* was equipped with long, plate-like endites similar to those of *Hurdia* and *Peytoia*, except that numerous moveable setae project from the anterior margin in place of stout auxiliary spines[19] (Fig. 3c). Each seta bears two rows of spinules (Fig. 3d), representing an additional order of complexity compared to the filtering apparatus of *Tamisiocaris*. With its long plate-like endites bearing anterior setae only, the

frontal appendage of *Pahvantia* is strongly reminiscent of that of *Aegirocassis*, except for an additional row of setae per endite, and apparent lack of spinules (Fig. 3e).

The highly complex filtering apparatus of the Ordovician taxon likely evolved in relation to gigantism, the presence of spinules allowing its mesh size to approach that of *Tamisiocaris* (490 µm), despite a huge difference in appendage size—a single endite of *Aegirocassis* is as long as the whole appendage of the early Cambrian species. Compared to the giant Moroccan species, *Pahvantia* has an inferred body length ten times smaller[20] and despite a simpler structure, a filtering system with a mesh size seven times smaller (c. 70 µm). This latter difference is important, for it suggests that *Pahvantia* was able to sieve much smaller organisms/particles out of the water column. Following Vinther et al.'s methodology[18], a minimum size between 70 µm (mesh size) and 100 µm (linear regression; Fig. 4) can be inferred for the suspended elements captured by the middle Cambrian radiodont. This corresponds to the size of large microplanktonic organisms, which nowadays essentially include autotrophic (phytoplankton) and heterotrophic (protozooplankton) unicellular eukaryotes[33] (Fig. 4). In other words, *Pahvantia* might have been both a primary and secondary consumer, and therefore contributed to a more efficient transfer of biomass and energy from the euphotic zone to benthic environments.

Indeed, even if relatively small for a radiodont (<25 cm), *Pahvantia* was a ten to thousand times larger than any mesoplanktonic (0.02–2 cm) primary consumers, and therefore

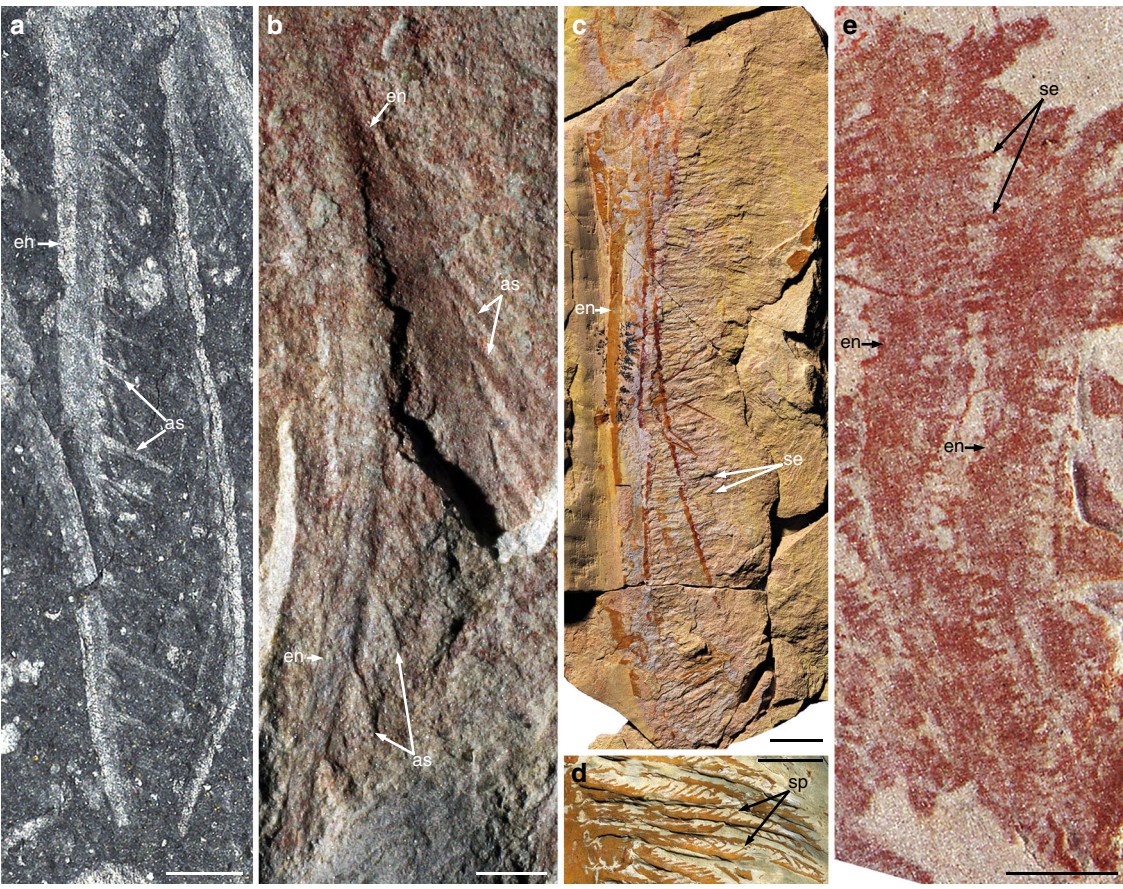

**Fig. 3** Endite morphologies in suspension-feeding radiodonts. Photographs of specimens immersed in dilute ethanol (**a**) or dry (**b**–**e**). **a**, **b** Endites with numerous auxiliary spines. **a** Early Cambrian *Tamisiocaris borealis* (MGUH30501; image: J. Vinther). **b** Early Cambrian *Anomalocaris briggsi* (SAM P47020a; image: J. Paterson). **c**, **d** Endite with numerous complex setae, Early Ordovician *Aegirocassis benmoulae* (images: P. Van Roy). **c** General view (YPM522227). **d** Detail of the setae and their spinules (YPM527125). **e** Endite with numerous simple setae, middle Cambrian *Pahvantia hastata* (KUMIP314089). Scale bars represent 1 cm (**c**, **d**) and 2 mm (**a**, **b**, **e**). as auxiliary spines, en endites, se setae, sp spinules

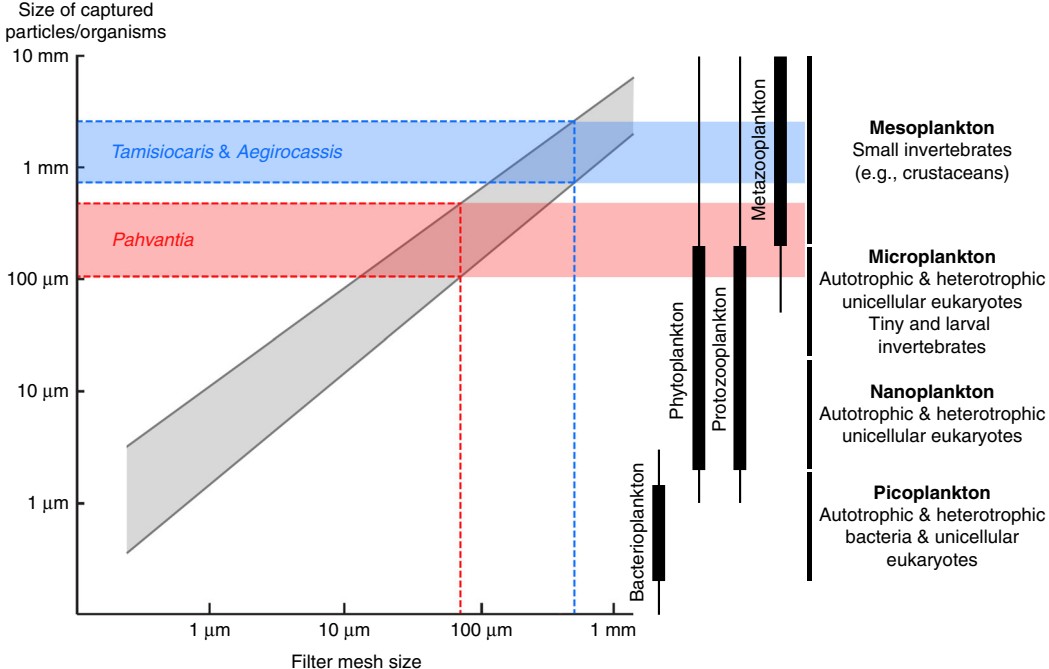

**Fig. 4** Size of the particles captured by suspension-feeding radiodonts. With a mesh size of 70 μm (red dashed line), *P. hastata* might have fed on microplankton, including phytoplankton. The mesh size was closed to 500 μm (blue dashed line) in both *Aegirocassis* and *Tamisiocaris*, which therefore likely grazed on mesoplankton. Grey area depicts prey-size variation for a given mesh size in modern suspension feeders. Diagram on the left is modified after ref. [18]. Size distribution of main plankton types (right) is derived from ref. [63]

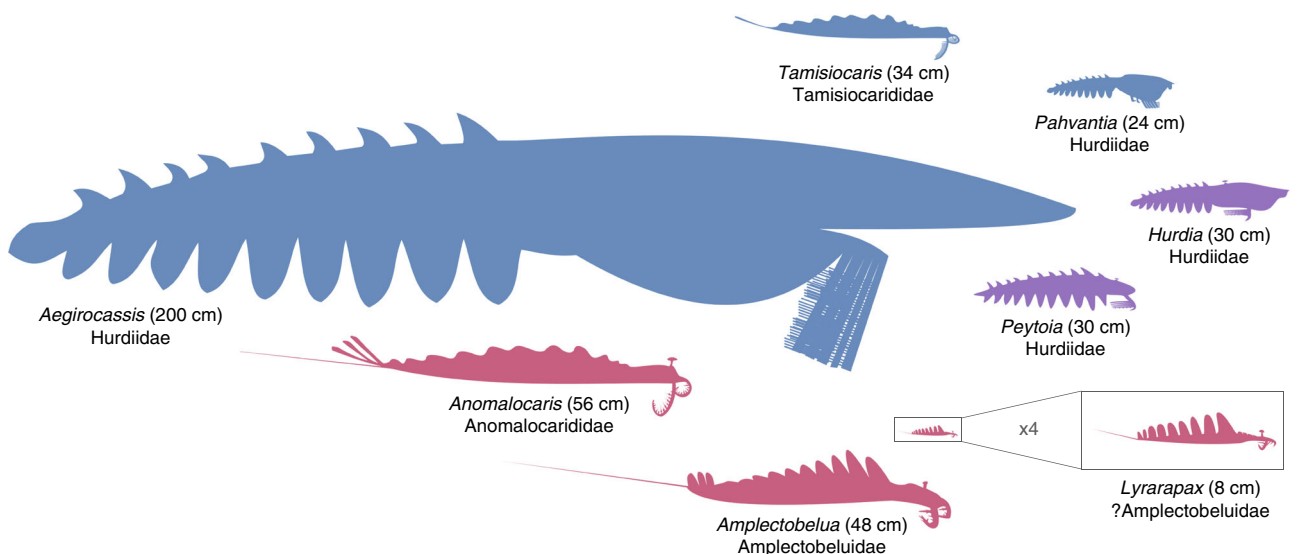

**Fig. 5** Diversity of sizes and feeding habits in radiodonts. As exemplified by *Aegirocassis* and *Pahvantia*, there is no obvious relationships between size and feeding strategy in this group. Raptorial predators are represented in red, sediment sifters in purple, and suspension feeders in blue

produced much larger and more rapidly sinking faecal pellets and carcasses. Larger organic remains have shorter residence times in the water column and consequently greater chances to reach the sea floor and become food for benthic organisms[34]. Thus, the large faecal pellets of *Pahvantia* were probably consumed by a variety of coprophagous animals, such as hyoliths, ptychopariid trilobites, and agnostoid arthropods[35], all abundant components of the Wheeler fauna[36]. Likewise, its carcasses likely supplied sustenance to mobile scavengers attracted from afar, including some arthropods (e.g. agnostoids[37,38]) and lobopodian and

scalidophoran worms[39,40]. Moreover, the efficiency of energy transfer between any two trophic levels is only 10% on average[41], and so by shortening the food chain connecting primary production and benthic organisms, *Pahvantia* might have more efficiently contributed to the flux of energy from pelagic to benthic realms than free-swimming animals feeding on mesoplankton (e.g. *Tamisiocaris*).

This hypothesis of a Cambrian nektonic organism like *Pahvantia* being an omnivorous suspension-feeder relies on the inferred minimum size of the particles trapped by its

filtering apparatus, and the fact that it falls within the size range of modern marine phytoplankton[33,42] (Fig. 4). This observation logically raises the question of whether phytoplanktonic organisms in Cambrian marine environments were comparable in size to modern ones. Unfortunately, little is known of the organisms responsible for marine primary production in the Early Palaeozoic, except that they were not coccolithophorids, diatoms or dinoflagellates, and that the role of green algae might have been greater than today[43]. Only the phytoplankton producing resistant cysts have left a trace in the fossil record in the forms of acritarchs. These organic-walled microfossils provide the only estimates of the sizes reached by primary producers at that time. According to Huntley et al.[44], average acritarch size dramatically increased during the Cambrian, reaching 55 µm during the 500–485 Ma time interval (i.e. shortly after the deposition of the Wheeler Formation). Extrapolating from the mean size (<10 µm) and size distribution of modern phytoplanktonic species[42], this average value for acritarchs suggests an upper size range of Drumian (c. 502 Ma) phytoplankton exceeding the mesh size (70 µm) of the filtering apparatus of *Pahvantia*. This seems all the more likely given that this estimated mesh size represents a maximum value, which only considers the distance between two adjacent setae of a given row. This assessment, measured in a single sub-adult specimen, omits the possibility that more distant elements (e.g. setae on adjacent endites) might have overlapped in the original three-dimensional organisation of the filtering basket, thus creating a finer filtering mesh[19]. Likewise, mesh size might have increased during the ontogeny of *Pahvantia*, as it does in some extant crustaceans[45,46]. Lastly, it cannot be entirely ruled out that spinules were present but not preserved in the middle Cambrian radiodont, which would considerably reduce the mesh size of the filtering net. In *Aegirocassis*, these structures are only conspicuous in a couple of particularly well-preserved specimens[19], and both are substantially much larger than the single, strongly flattened *Pahvantia* appendage described herein. Considering these possible biases, it is particularly likely that this macroscopic nektonic organism grazed on large phytoplankton, especially during the early phases of its life.

The description of a Drumian suspension-feeding radiodont illustrates that this feeding strategy evolved at least twice, possibly three times, in the history of this group: *Tamisiocaris* (early Cambrian), *Pahvantia* (middle Cambrian), and *Aegirocassis* (Early Ordovician). The three taxa are all retrieved within a clade regrouping Tamisiocarididae and Hurdiidae, with *Tamisiocaris* belonging to the former and *Aegirocassis* and *Pahvantia* to the latter. An ancestor-descendant relationship between *Pahvantia* and *Aegirocassis* seems unlikely according to the association of the latter taxon with *Peytoia* in some of the retrieved trees of the parsimony analyses. This polyphyletic origin of filter-feeding in radiodonts mirrors the situation observed in recent crustaceans, where this feeding mode is known in representatives of at least fifteen orders[46]. This reflects the fact that microscopic suspension-feeding and macroscopic raptorial predation are two extremes of the same spectrum, which essentially differ in the size of the particles/organisms caught, rather than in the mechanics of feeding. This is well-illustrated by extant crustaceans with complex filtering apparatuses, which nonetheless commonly (e.g. krills[47,48]; copepods[49]) or occasionally (e.g. barnacles[50–53]) engage in raptorial predation, or by the progressive shift from filter-feeding to raptorial predation during ontogeny in some fairy shrimps[54], barnacles[52], or krills[47]. Actually, there might be a positive correlation between body size and contribution of animal-derived food to the diet in crustaceans, with larger taxa/individuals preying more and targeting larger preys. This led Riisgård[46] to hypothesize the existence of a physiological limitation to body size in filter-feeding crustaceans, especially herbivorous ones. In short, feeding in these organisms is constrained by the size of the filtering surface, while metabolism (especially respiration) is related to body volume; a surface increasing slower than a volume, there should be a maximum body size beyond which suspension-feeding cannot compensate metabolic cost. The discovery of a 200 cm-long, filter-feeding Ordovician radiodont indicates that the relationship between body size and feeding strategy was more complicated in these extinct organisms. Vinther et al.'s suggestion[18] – somewhat reverse to Riisgård's hypothesis—that suspension feeding animals evolved from large predators might also be nuanced. Indeed, albeit reaching a large body size compared to most middle Cambrian animals, *Pahvantia* remains one of the smallest radiodonts[20] (Fig. 5). Moreover, the sizes of the presently-known filter-feeding radiodonts positively correlate with phytoplankton (acritarch) diversity[55,56], which suggests that food availability, more than ancestor size, might have governed the sizes reached by these organisms.

Thus, there are no clear relationships between body size, feeding strategies, and phylogenetic affinities in radiodonts (Fig. 5), which actually attests to the great adaptability of these extinct organisms. This is particularly well-illustrated by the wide range of morphologies and functions displayed by their frontal appendages[25,27,31]. The description of a microplanktonic-feeder representative adds to an ever-growing body of evidence suggesting that radiodonts, as both juveniles and adults[27], contributed in various ways to the emergence of modern-style marine ecosystems in the Cambrian, including in the coupling of pelagic and benthic realms.

## Methods

**Fossil illustration**. After being delicately prepared using a needle, the new specimens of *P. hastata* (KUMIP 134187, 314089, 314090) were photographed dry or immersed in dilute ethanol using a Leica DFC420 digital camera mounted on a Leica MZ16 microscope, with manual focusing at different focal planes and subsequent stacking in Adobe Photoshop CS6; this software was also used to prepare the figures.

**Phylogenetic analysis**. To explore the phylogenetic position of *Pahvantia* (Fig. 2 and Supplementary Figs 4, 5) and the impact of its characters on the phylogeny of radiodonts, we conducted a phylogenetic analysis using a significantly modified version of the character matrix used in Van Roy et al.[19], itself derived from Vinther et al.[18]. Composed of 61 morphological characters and 33 taxa[20] (Supplementary Note 2), this dataset was analysed through parsimony inference with TNT[57] using New Technology Search (Driven Search with Sectorial Search, Ratchet, Drift, and Tree fusing options activated) in standard settings under equal and implied weighting[58,59]. The analysis was set to find the minimum tree length 100 times and to collapse trees after each search, and all characters were treated as unordered. Additionally, we subjected the same dataset to a Bayesian inference analysis run in MrBayes 3.2 using the Monte Carlo Markov-chain (MCMC) model for discrete morphological characters[60,61] for 5 million generations (four chains), with every 1000th sample stored (resulting in 5000 samples), and 25% burn-in (resulting in 4500 retained samples). Convergence was verified when effective sample size (ESS) values were over 200 for all parameters, and corroborated graphically using the software Tracer v.1.6[62].

## Data availability

The data generated or analysed during this study are included in the published article (and its supplementary information files) or available in Dryad Data Repository with the identifier https://doi.org/10.5061/dryad.1cf2fb0 (2018)[20]. Photographic material of the studied fossils is available from the corresponding author upon request. The specimens of *A. benmoulai*, *A. briggsi*, *P. hastata*, and *T. borealis* illustrated herein are housed in the collections of the Yale Peabody Museum of Natural History (USA; YPM), the South Australian Museum (Australia; SAM), the University of Kansas Natural History Museum (USA; KUMIP), and the Geological Museum of the University of Copenhagen (Denmark; MGUH), respectively.

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

## Acknowledgements

We are particularly grateful to Glade Gunther, Lloyd Gunther, Val Gunther, and R.L. Harris, who found the specimens of *Pahvantia hastata* and generously donated them to the Kansas University Museum of Invertebrate Palaeontology, as well as B. Lieberman and especially J. Kimmig for making them available to us for the study. We thank J. Kimmig, J. Paterson, P. Van Roy, and J. Vinther for providing us with the images in Supplementary Figure 2a–d (*P. hastata*), Fig. 3b (*A. briggsi*), Fig. 3c and d (*A. benmoulae*), and Fig. 3a (*T. borealis*), respectively. J. Ortega-Hernández and P. Van Roy kindly helped with the cladistic analysis. S.P. was supported by an Oxford-St Catherine's Brade-Natural Motion Scholarship.

## Author contributions

R.L.-A. conceived the project, prepared and photographed the new specimens, and conducted the phylogenetic analysis with inputs from S.P. The latter compiled the data related to the body lengths of radiodonts. R.L.-A. wrote the manuscript with inputs from S.P., who prepared Fig. 5; R.L.-A. prepared the other figures.

## Additional information

**Competing interests:** The authors declare no competing interests.

