## [Peer Review File · Nature Communications]

Reviewers' comments:

Reviewer #1 (Remarks to the Author):

This is a well written and illustrated manuscript that develops the diversity of the feeding modes amongst the radiodonts. A useful contribution providing mechanisms for nutrient transfer from the water column to the seabed as a possible biological pump for the Cambrian Explosion. I have made more minor comments on the MS.

Prior to Vinther et al. 2014 it was difficult to persuade the community that the radiodonts were anything other than apex predators. There is now a diversity of trophic modes identified within the group. This paper is attractive (and original) for two main reasons. It builds on the the matrix approach of Vinther et al. to identify the size of particles consumed and thus indicates that this form targeted smaller organisms and particles (these were thus by implication probably quite abundant in the water column) and secondly it provides a nutrient transfer mechanism to the seabed and thus the benthos. However the authors might like to comment on the widespread idea that the Cambrian seas were actually dominated by nekobenthic and nektonic predators; what types of benthic organisms would have consumed and recycled the products of this biological pump.

The phylogenetic analyses are carefully executed and they develop the position of the sweep-net, suspension feeders on the radiodont tree.

The authors might wish to give a bit more background to the Cambrian Explosion (Smith and Harper 2013) and its extent (Daley et al. 2018).

Reviewer #2 (Remarks to the Author):

This manuscript identifies a new filter-feeding radiodont, *Pahvantia hastata*, from the middle Cambrian of Utah. The tripartite cephalic carapace and the elongated endites of the frontal appendage unequivocally indicate that *P. hastata* is a member of the family Hurdiidae, a largest family of radiodonts. Intriguingly, some endites (on the frontal appendage) of this hurdiid taxon bear a pair of rows of setae, which are only spaced 60-70 μm with each other. Those elegant setae are interpreted as functioning to filter food particles of a size equivalent to that of microplankton (and mesoplankton). This new discovery shows that Cambrian suspension-feeding animals, notably radiodonts (traditionally regarded as top predator in Cambrian sea), could transfer the energy produced by variously sized primary producers, thus must have played important roles in establishing the earliest oceanic pelagic-benthic energy flux. The study provides further key evidence on the establishment of oceanic trophic chain and reveals the complexity of the earliest marine ecosystem dominated by metazoans.

The results are of significance and of wide interest. Data are very well described and documented, and the interpretation is convincing. The text is well written, and the references are well selected.

Some questions:

- line 84 p. 4. 'a pair of deep lateral elements'. It is weird to describe carapace as 'deep', need further explain for this.
- line 96-97 p. 7. The peduncle of the frontal appendage is preserved as 'particularly narrow'. This might be a taphonomic result rather than true biological feature, as the (relatively) long and large distal part of the frontal appendage would require a robust peduncle. Given the preservation of the key specimens and the irregular edge of the peduncle, it is not convincing to conclude that the peduncle is narrow.
- The phylogenetic result is not good enough in terms of the resolution of (and within) hurdiids and 'cetiocaridids', which include the three known radiodonts having suspension-feeding habit. To argue that suspension-feeding in different group of radiodonts might have evolved multiple times, a robust phylogenetic tree is essential. Recently, some people have been arguing that the Bayes-based method might perform better than the parsimony-based cladistic analysis. It would be worth to try a Bayes analysis on current dataset.

With these few comments I would suggest to accept the manuscript after minor revision.

Point-by-point response to the referees' comments:

REVIEWER #1 (anonymous)

1) *I have made more minor comments on the MS.*

Changes: All the changes suggested by Referee 1 directly on the MS were followed.

2) *The authors might like to comment on the widespread idea that the Cambrian seas were actually dominated by nektobenthic and nektonic predators?*

Reply: Cambrian classical ('shelly') fossil assemblages are dominated by trilobites, hyolithids and brachiopods. Some trilobites might have been predators/scavengers, but most of them were more likely particle feeders (Lerosey-Aubril & Peel, 2018). This is particularly true of the Ptychopariida, the most diverse trilobites in the Drumian (i.e. the time when *Pahvantia* lived), for they do not show any of the morphological features generally associated with predatory habits (e.g. conterminant/impudent hypostomal condition). Hyolithids are thought to have been suspension-feeders (Moysiuk et al. 2017) or coprophagous (Kimmig & Pratt 2018), and brachiopods were/are all suspension-feeders.

If we consider exceptionally-preserved biotas, these are typically dominated by arthropods, sponges, and brachiopods. Arthropods are extremely varied in feeding habits, from predators/scavengers to filter-feeders to deposit/detritus-feeders. Moreover, many of the putative arthropod predators from Cambrian Konservat-Lagerstätten have been regarded as predators because they possessed digestive glands, but the presence of such organs is not restricted to predators in modern arthropods and therefore cannot be used alone to infer predatory habits in extinct taxa. Sponges are suspension-feeders, as are brachiopods.

Quantitative studies of Cambrian remarkable fossil assemblages, such as those of the Greater Phyllopod Bed or the Tulip Beds (both from the Burgess Shale), actually contradict the claim that predators dominated these communities. In the Greater Phyllopod Bed, either suspension feeders or deposit feeders dominate in all beds (Caron & Jackson, 2008, fig. 3); in the Tulip Beds, fossil assemblages are dominated by suspension feeders in term of diversity, and deposit feeders in term of abundance (O'Brien & Caron, 2016, figure 8).

Changes: Because we could not find support for this particular comment of referee 1, we made no changes to our text.

3) *What types of benthic organisms would have consumed and recycled the products of this biological pump?*

Reply: As a relatively large (30 cm long) nektonic animal, *Pahvantia* contributed to the transfer of organic matter from pelagic to benthic realms through the production of large faecal pellets and carcasses. The former were likely consumed by coprophagous animals (e.g. hyolithids) and the latter by scavengers (e.g. agnostoid arthropods). The Wheeler fauna comprises many organisms commonly associated with one or the other of these two feeding strategies.

Changes: The following two sentences were added to the discussion:

“Thus, the large faecal pellets of *Pahvantia* were probably consumed by a variety of coprophagous animals, such as hyoliths, ptychopariid trilobites, and agnostoid arthropods (Kimmig & Pratt, 2018), all abundant components of the Wheeler fauna (Robison et al. 2015). Likewise, its carcasses likely supplied sustenance to mobile scavengers attracted from afar, including some arthropods (e.g. agnostoids; Chatterton et al. 2003; Zaccai et al. 2016) and lobopodian and scalidophoran worms (Vannier, 2012; Vannier & Martin, 2017).”

The six new references were added to the reference list and all the references cited after them in the text were renumbered.

4) *The authors might wish to give a bit more background to the Cambrian Explosion (Smith and Harper 2013) and its extent (Daley et al. 2018).*

Reply: We agree with reviewer 1.

Changes: We added some information on the Cambrian Explosion (definition, duration, impact) in our introduction:

“The early Cambrian Period is associated with an even greater **modification of marine life** – the advent of metazoans and their organisation in complex ecosystems. **In less than 30 Myr, most major bilaterian phyla appeared and greatly diversified (Erwin et al., 2011; Daley et al. 2018), provoking a dramatic change in both the composition and the functioning of the biosphere, which in turn profoundly and irreversibly impacted the Earth system (Butterfield, 2007, 2011). For instance, likely triggered by a new ecological force – predation – some of these early Cambrian animals (e.g. scalidophorans) rapidly evolved aptitudes for life within soft substrate (Buatois et al. 2016), thus increasing the breadth and depth of eukaryotic colonization of the sea floor, and concomitantly the habitability of the latter (Mangano & Buatois, 2014).”**

The five new references were added to the reference list and all the references cited after them in the text were renumbered.

REVIEWER #2 (anonymous)

1) *Line 84, p. 4: 'a pair of deep lateral elements'. It is weird to describe carapace as 'deep', need further explain for this.*

Reply: The term 'deep' was initially used following the authoritative contribution of Daley et al. (2013) on the Burgess Shale genus *Hurdia* (type genus of the family Hurdiidae). Yet, it is true that the vertical dimension of a skeletal element is more commonly described as its height.

Changes: 'Deep' was replaced by 'tall'.

2) *Line 96-97, p. 7: The peduncle of the frontal appendage is preserved as 'particularly narrow'. This might be a taphonomic result rather than true biological feature, as the (relatively) long and large distal part of the frontal appendage would require a robust peduncle. Given the preservation of the key specimens and the irregular edge of the peduncle, it is not convincing to conclude that the peduncle is narrow.*

Reply: We agree with referee 2 that the peduncle in the sole specimen displaying this structure might be incompletely and/or obliquely preserved. As we wrote in the sentence immediately before, some aspects of the morphology of this appendage are not completely understood due to preservation.

Changes: In the main text, the peduncle is only briefly mentioned in the part describing the fossil illustrated in Figure 1. This text describes the appendage as it is preserved, not as it might have been in the living organism and therefore, we only added a few words at the end of the sentence:

"The peduncle (proximal part) is particularly narrow proximally and widens distally (Fig. 1c, e), possibly indicating an oblique orientation relative to bedding **and/or incomplete preservation.**"

In addition, we added a sentence to Supplementary Note 1, which responds to referee 2's comment in more details:

"For instance, the peduncle appears abnormally slender for a structure aiming to support to the distal region of the appendage and its large endites – this could be explained by an oblique orientation and/or an incomplete preservation of this proximal region. The morphology of the setae-bearing endites and their apparent branching into two parts distally are even more complicated to understand."

3) *The phylogenetic result is not good enough in terms of the resolution of (and within) hurdiids and 'cetiocaridids', which include the three known radiodonts having suspension-feeding habit. To argue that suspension-feeding in different group of radiodonts might have evolved multiple times, a robust phylogenetic tree is essential. Recently, some people have been arguing that the Bayes-based method might perform better than the parsimony-based cladistic analysis. It would be worth to try a Bayes analysis on current dataset.*

Reply: We agree with Reviewer 2 that the resolution of the trees resulting from our phylogenetic analysis (and all previously published ones) is rather low. However, we clearly show in our manuscript that the topology illustrated by the strict consensus tree is fundamentally preserved in all the trees obtained using equal or implied weighting. Importantly, in all these trees *Pahvantia* is recovered within a monophyletic family Hurdiidae, to which *Tamisiocaris* never belongs – the latter taxon typically forms a monophyletic family 'Cetiocarididae' with *A. briggsi*. This indicates that suspension-feeding evolved independently twice (at least) in radiodonts. What remains unclear is whether this feeding strategy evolved twice within the Hurdiidae (and so three times in Radiodonta as a whole), for the relationships between *Aegirocassis* and *Pahvantia* are not sufficiently well-resolved yet. Moreover, Bayesian inference analyses may or may not provide more reliable results than parsimony analyses (e.g. Samson et al. 2018), and in most cases they produce lesser-resolved trees.

Nevertheless, we agree with Reviewer 2 that a Bayesian analysis has the potential to be informative. Thus we followed Referee 2's recommendation and subjected our dataset to a Bayesian inference analysis. As expected, the resulting tree has a lower resolution than the ones obtained with parsimony analysis (the radiodont main clades are in a polytomy with deuteropods), but once again it clearly shows that *Pahvantia* belongs to a monophyletic Hurdiidae, while *Tamisiocaris* and *A. briggsi* form a separate clade, 'Cetiocarididae'. These results provide further support to the claim that suspension-feeding evolves twice in radiodonts, and we have included them in our revised manuscript.

Changes:

1. We added the following paragraph to the section 'Results' (subsection 'Phylogenetic relationships'):

"The same dataset was also subjected to Bayesian inference analysis (Supplementary Fig. 5), which recovers Pahvantia as part of a monophyletic Hurdiidae. 'Cetiocarididae' and the core group

Amplectobeluidae (Amplectobelua, Lyrarapax and Anomalocaris kunmingensis) also form monophyletic groups. All the other radiodonts are retrieved in polytomy with these three clades and a monophyletic Deuteropoda, except for Caryosyntrips that occupies a more basal position. Despite a general loss of resolution inherent to Bayesian inference analysis (O'Reilly et al. 2016), the topology of this tree is fundamentally congruent to those obtained using parsimony analysis (Supplementary Figure 4).

Whatever the methods used, the resulting trees clearly illustrate two important points: 1) Pahvantia belongs to the Hurdiidae and 2) suspension-feeding evolved in at least two independent lineages of radiodonts ('cetiocaridids' and hurdiids)."

2. The following sentences were also added in the Methods section:

"Additionally, we subjected the same dataset to a Bayesian inference analysis run in MrBayes 3.2 using the Monte Carlo Markov-chain (MCMC) model for discrete morphological characters (Lewis, 2001; Ronquist et al., 2012) for 5 million generations (four chains), with every 1000th sample stored (resulting in 5000 samples), and 25% burn-in (resulting in 4500 retained samples). Convergence was verified when effective sample size (ESS) values were over 200 for all parameters, and corroborated graphically using the software Tracer v.1.6 (Rambaut, A. et al. 2017)."

3. Four new references were added to the reference list and the numbering of the in-text citations has been modified accordingly.
4. A new Supplementary Figure 5 has been created to illustrate the results from the Bayesian Inference analysis.

REFERENCES

- Caron, J. B. & Jackson, D. A. Paleocology of the Greater Phyllopod Bed community, Burgess Shale. *Palaeogeogr. Palaeoclimat. Palaeoecol.* **258**, 222–256 (2008).
- Daley, A. C., Budd, G. E. & Caron, J.-B. Morphology and systematics of the anomalocaridid arthropod *Hurdia* from the middle Cambrian of British Columbia and Utah. *J. Syst. Palaeontol.* **11**, 743–787 (2013).
- Kimmig, J. & Pratt, B. R. Coprolites in the Ravens Throat River Lagerstätte of Northwestern Canada: Implications for the Middle Cambrian Food Web. *Palaios* **33**, 125–140 (2018).
- Lerosey-Aubril, R. & Peel, J. S. Gut evolution in early Cambrian trilobites and the origin of predation on infaunal macroinvertebrates: evidence from muscle scars in *Mesolenellus*. *Palaeontology* (2018). DOI: 10.1111/pala.12365
- Moysiuk, J., Smith, M. R. & Caron, J. B. Hyoliths are Palaeozoic lophophorates. *Nature* **541**, 394–397 (2017).
- O'Brien, L. J. & Caron, J. B. Paleocommunity analysis of the Burgess Shale Tulip Beds, Mount Stephen, British Columbia: comparison with the Walcott Quarry and implications for community variation in the Burgess Shale. *Paleobiology* **42**, 27–53 (2016).
- Sansom, R. S., Choate, P. G., Keating, J. N. & Randle, E. Parsimony, not Bayesian analysis, recovers more stratigraphically congruent phylogenetic trees. *Biology Letters* **14**, 20180263 (2018). <http://dx.doi.org/10.1098/rsbl.2018.0263>

REVIEWERS' COMMENTS:

Reviewer #2 (Remarks to the Author):

The authors have addressed all the questions I have asked in the last round of reviewing. Their response is satisficing and I delightly to say that this paper is ready for publication.